# Rift and plume: a discussion on active and passive rifting mechanisms in the Afro-Arabian rift based on synthesis of geophysical data

Ran Issachar[1,2], Peter Haas[1,3], Nico Augustin[3] and Jörg Ebbing[1]

[1]Institute for Geosciences, Kiel University, Geophysics, Kiel, Deutschland, [2]Geological Survey of Israel, Jerusalem, Israel, [3]GEOMAR Helmholtz Centre for Ocean Research, Kiel, Deutschland

*Correspondence to*: Ran Issachar (ranis@gsi.gov.il)

## Abstract

The causal relationship between the activity of mantle plumes and continental break-up is still elusive. The Afro-Arabian rift system offers an opportunity to examine these relationships, in which an ongoing continental break-up intersects a large Cenozoic plume-related flood basalt series. In the Afar region, the Gulf of Aden, the Red Sea, and the Main Ethiopian Rift form an R-R-R triple junction within plume related flood basalts series. We provide an up-to-date synthesis of the available geophysical and geological data from this region. We map the rift architecture in the intersection region by applying the~~using~~ Difference in Gaussians to the topography and the bathymetry and interpreting ~~interpretation of~~ vertical gravity gradients and Bouguer anomalies. With the aid of these methods we review the spatio-temporal constraints in the evolution of the different features of the plume-rift system.

Our results show rough and irregular morphologies ~~morphology~~ of the Gulf of Aden and the Red Sea arms in contrast to the symmetric, continuous, and smooth Main Ethiopian Rift. The triple junction formed by the northeastward propagation of the Main Ethiopian rift ~~develops~~ and developed simultaneously to the abandonment of the tectonic connection between the Red Sea and the Gulf of Aden through Bab al-Mandab Strait. The onset of the triple junction was the last feature to develop in the plume-rift system and marked a tectonic reorganization. By this time, all rift arms were sufficiently evolved and the break-up between Africa and Arabia was already accomplished.

We argue that the classical active and passive rifting mechanisms cannot simply explain the progressive development of the Afro-Arabian rift. Instead, we propose a plume-induced plate rotation, which includes an interaction between active and passive mechanisms. In this tectonic scenario, the arrival of the Afar plume provided a push force that promoted the rotation of Arabia around a nearby pole located to the northwest ~~to~~ of the plate boundary, enabling the rifting and, ultimately, the break-up of Arabia from Africa.

Short summary:

In this contribution, we explore the causal relationship between the arrival of the Afar plume and the initiation of the Afro-Arabian rift. We mapped the rift architecture in the triple junction region using geophysical data and reviewed the available geological data. We interpret a progressive development of the plume-rift system and suggest an interaction between active and passive mechanisms in which the plume provided a push-force that changed the kinematics of the associated plates.

# 1. Introduction

The causal dependency between the eruption of flood basalts and continental break-up is still unclear, although a close occurrence between these two phenomena has been recognized for a long time. Continental flood basalts, often referred to as traps, form large igneous provinces covering huge continental areas (Bryan and Ferrari, 2013; Ernst, 2014). Continental flood basalts are often associated with extensive volcanism during short time intervals, which are brought to the surface by deep-seated mantle plumes (Richards et al., 1989; White and McKenzie, 1995; Koppers et al., 2021), although other mechanisms were also suggested (e.g., Anderson, 1994, 2005). There is evidence for a close temporal and spatial occurrence between the eruption of flood basalts and continental break-up. In particular, when reconstructed back to their original plate tectonic configuration, a R-R-R triple junction is typically found within the flood basalt areas (Morgan, 1971; Burke and Dewey, 1973; Buiter and Torsvik, 2014). Using the geological record to examine the mutual dependency of these processes is challenging. It requires high-precision in the temporal and spatial development of the volcanic and tectonic features, often obscured by the overprint of different tectonic processes.

The Afar region in the central parts of the Afro-Arabian rift system is recognized as a key locality to examine models of plume-rift association, offering a young and active case study in which a plume, regional uplift, an R-R-R triple junction, break-up, and oceanic spreading co-exist and are superimposed (Fig. 1). Plume-rift association is mainly explained as either 'active' (e.g., Sengör and Burke, 1978) or 'passive' (e.g., White and McKenzie, 1989), with no interaction between those mechanisms. Despite the contrary implications, the Afar case study was used as a prime example to support both the 'active' (e.g., Burke and Dewey, 1973) and the 'passive' (e.g., White and McKenzie, 1989) mechanisms, and some authors argued that both processes are required to explain the observations (e.g., Courtillot et al., 1999). The discrepancy can be primarily attributed to the lack of accurate geological and geophysical evidence regarding the uplift, volcanism and rifting phases. Moreover, ~~detailed compression between~~studies show that changes in plate motions and the activity of plumes occurred concurrently, suggest~~s~~ new concepts in which plumes cause rapid deviations in the kinematics of nearby plates (e.g., Cande and Stegman, 2011).

The purpose of this paper is to discuss the causal relationship between the Afar plume and rifting along the Afro-Arabian rift system in light of the large amounts of data collected in recent years and the new concepts derived from studies of the Indian plate~~other case studies~~. For this, we first review the timing of volcanism and uplifts in Ethiopia and Yemen, and~~,~~ the timing of rifting along the Gulf of Aden, the Red Sea and the Main Ethiopian rift. We further provide an analysis and interpretation of modern geophysical datasets, including topography, bathymetry~~,~~ and gravity, and incorporated the distribution of offshore magnetic anomalies, earthquakes (ML>4), and onshore Quaternary volcano~~s~~ ~~distribution~~. Using these datasets, we map the architecture of the rifts and describe the development of rift segments. Finally, we compare our results with recent models and other case studies in the world, aiming to shed light on the causal relationship between mantle plumes and tectonic processes.

# 2. Active and passive mechanisms for plume-rift association

The existence of deep mantle convection and its interaction with the Earth's lithosphere was already pointed out by Wilson (1963), and a close occurrence to continental break-up was soon noticed by the

abundance of hotspots near many rift junctions (Morgan, 1971) and flood basalt volcanism along passive
margins (Richards et al., 1989). Although Morgan (1971) speculated that deep mantle convection has a
significant role in accelerating the overlying tectonic plates, it was later realized that slab-pull provides the
main driving force for plate motion (Forsyth and Uyeda, 1975). In their landmark paper, Burke and Dewey
(1973) presented 45 case studies of rift junctions associated with hot spots. They proposed a model in
which plume-associated uplift and volcanism precede and generate the rift arms, initiated from a triple
junction within the plume region. Afar was used as a first and prime example, highlighting its importance
as a young and active case study; however, they already noted a complex distribution of continental
fragments and magnetic anomalies (Burke and Dewey, 1973).
Following these insights, 'active' rifting models were developed to explain plume-rift associations (e.g.,
Keen, 1985; Moretti and Froidevaux, 1986; Campbell and Griffiths, 1990; Hill, 1991; White and McKenzie,
1995). These models generally propose that rifting can result from a combination of processes derived
from the actively rising head of an anomalously hot mantle. These mantle plumes impinge and erode the
base of the lithosphere, which prompt uplift and decompression melting,  introduce internal extensional
forces and ultimately lead to break-up. Accordingly, regional uplift and volcanism are expected to precede
rifting, which would initiate from a triple junction above the mantle plume head (Fig. 2a).
Later contributions challenged the active view, arguing that a 'passive' asthenospheric upwelling can also
resolve the occurrence of flood basalt near rifts (firstly introduced by White and McKenzie, 1989). In this
view, rifting is initiated by the remote extensional stresses, usually along former sutures and weak zones,
regardless of underlying plumes. The production of massive volcanism is allowed when the thinned and
stretched lithosphere is underlaid by a thermal anomaly in the mantle. The volcanism is generated by
decompression melting of the hot asthenospheric mantle, which passively rises. As plumes form large
areas of high temperatures in the mantle, massive volcanism is found on Earth's crust close to rifts.
Accordingly, subsidence is a precondition required for magmatism, and there is no triggering mechanisms
for a triple junction to form within the flood basalts region (Fig. 2b).
Although active and passive mechanisms have been discussed in the last 50 years, the role of plumes in
initiating rifting is still unclear and much debated. Even for well-studied and prime examples of plume-rift
association as the Siberian, Parana-Etendeka, Deccan, and Greenland traps, there is no agreement on
whether active processes initiated rifting (Geoffroy, 2005; Ivanov et al., 2015; Frizon De Lamotte et al.,
2015; Fromm et al., 2015; Mitra et al., 2017). Some authors emphasize the significance of pre-existing
lithosphere weaknesses along former sutures and structures (Buiter and Torsvik, 2014; Will and Frimmel,
2018), while others show the potential of plumes to thermally and chemically erode the base of the
lithosphere in the weakening process allowing rifting (Sobolev et al., 2011). Additionally, some models
demonstrate that mixed active-passive scenarios can better explain observation (Koptev et al., 2018), and
even that both mechanisms are needed to explain temporal variations in rifts (Huismans et al., 2001).
In addition to the dichotomic views, a complex relationship in which plumes can influence the horizontal
velocities of plates is suggested based on detailed plate reconstructions and numerical modeling (van
Hinsbergen et al., 2011, 2021; Cande and Stegman, 2011; Chatterjee et al., 2013; Pusok and Stegman,
2020). In these studies, an abrupt changes in plate velocities is correlated to the arrival of a nearby plume
head. In the kinematic record of the Indian plate, the arrival of the Marion and Reunion plumes (associated
with the Morondava and Deccan LIPs) is synchronized with an abrupt plate speed-up and an Euler pole
shifting. During the arrival of the Reunion plume (at ~65 Ma), the acceleration of the Indian plate was
coupled with transitory slowing of the African plate (Cande and Stegman, 2011). Plume push forces,
sourced by the drag of the flowing asthenosphere, werewas shown as capable to change the plate
kinematics of nearby plates and even trigger the formation of new plate boundaries by a mechanism
termed as plume-induced plate rotation (van Hinsbergen et al., 2021) (Fig. 2c).

# 3. Geological setting

The Afro-Arabian rift system extends from Turkey to Mozambique (McConnell and Baker, 1970) and is the
current episode of the Phanerozoic break-up of the East African continental plate (Bosworth, 2015). It
contains rifting in the Gulf of Aden, in the Red Sea, and in East Africa. In the center of that system, the
Ethiopian northwestern and southeastern plateaus represent an elevated topography with a highest peak
of 4,620 m (Ras Dashan) and an average elevation of 2000 m above sea level. This area is part of the so-
called African Superswell, a wide region of anomalously high topography comprising East Africa (Lithgow-
Bertelloni and Silver, 1998; Corti, 2009). In western Yemen, the Sarawat Mountains are the highest peaks
in the Arabian Peninsula, reaching more than 3,000 m, at only 100 km distance away from the shoreline
of the Red Sea. These mountains show a typical stair morphology with steep slopes at the western and
southern sides, while the eastern shows gentler downward slopes.
The Gulf of Aden is the most developed rift segment in the Afro-Arabian rift, with a mature and fully
developed oceanic spreading center connected to the mid-ocean ridge in the Indian Ocean. Six pairs of
magnetic anomalies associated with seafloor spreading are recognized along the Gulf of Aden (Fournier et
al., 2010) (Fig. 3). Oblique rifting and high-angle structural inheritance along the Gulf of Aden resulted in
multiple ridge segments and fracture zones (i.e., transform faults; Leroy et al., 2013; Autin et al., 2013;
Bellahsen et al., 2013; Duclaux et al., 2020).
At the northern parts, the rifting in the Red Sea is connected by the Dead Sea Fault to the Eurasian collision
zone along the Taurus-Zagros Mountains. The Red Sea is experiencing the last stages of break-up and early
stages of oceanic accretion. An oceanic spreading center with three pairs of ridge parallel magnetic
anomalies are recognized in the southern parts of the Red Sea (Schettino et al., 2016) (Fig. 3)., Hhowever,
oceanic crust is probably flooring most of the basin (Augustin et al., 2021).
The Main Ethiopian Rift is the northernmost section of the intra-continental rifting in East Africa, splitting
the not-yet well-individualized Somali plate from Africa (Chorowicz, 2005). Current Active rifting in the
Main Ethiopian Rift is characterized by a narrow rift valley, in which volcanic and tectonic activities are
localized and influenced by oblique rifting (Corti, 2009).
The above-mentioned three rift arms meet in the Afar triangle (Fig. 3). It is a low elevated area compared
to the high Ethiopian plateaus and thus commonly referred to as the Afar 'depression'. Nevertheless, this
term is misleading as the Afar triangle is included within the rifted area and is geologically elevated from
the deep bathymetry of the Gulf of Aden and the Red Sea basins. The Afar triangle is mainly floored by
Pliocene and younger volcanic rocks, where Miocene volcanic series are exposed along the western
margins and at the elevated Danakil block. It comprises many volcanoes and axial volcanic ranges (Fig. 2),
where the northeastern side is characterized by transverse volcanic fields and the southwestern side by
central volcanoes (Varet, 2018). Two symmetric magnetic anomalies have been recognized in the Tendaho
graben, similar to those observed along spreading centers in the Gulf of Aden (Bridges et al., 2012). These
could be associated either with young oceanization or with linear anomalies developed in transitional crust
(Ebinger et al., 2017). Structurally, several mega-scale accommodation zones are connecting the different
rift segments and the location of a main triple junction location areis recognized at 11.0°N, 41.6°E atalong
the Tendaho-Goba'ad Discontinuity (e.g, Tesfaye et al., 2003) (Fig. 3).

# 4. Temporal constraints

### 4.1. Flood basalts and uplift

Vast efforts were made to study the chemistry and chronology of flood basalts in East Africa (see review
by Rooney, 2017). Two phases of extensive flood basalt volcanism are associated with plume-lithosphere
interaction (Fig. 4). The early phase is mainly confined to southern Ethiopia and northern Kenya. The timing
of this event is poorly constrained to 45-35 Ma (George et al., 1998). The second phase of flood basalt
eruptions was more voluminous, more widespread, and shorter-lived.  Earliest basalts of this phase date
back to 34 Ma near the Tana Basin in Ethiopia (Prave et al., 2016) and 31 Ma in western Yemen (Peate et
al., 2005) (Fig. 4). The traps accumulated very rapidly, in less than 6 Ma (Coulié et al., 2003), and include
tholeiitic to alkaline compositions of asthenosphere mantle source (Mattash et al., 2013). Thick sequences
of up to 2 km are observed within a widespread region in Ethiopia and Kenya (Bellieni et al., 1981; Wescott
et al., 1999; McDougall and Brown, 2009). It is commonly accepted that these flood basalts are of a deep-
seated mantle plume origin (Koppers et al., 2021). However, the formation mechanism is debatable and
may involve multiple plume impingements within a broad upwelling zone connected to the African
superplume in the lower mantle (Meshesha and Shinjo, 2008) or a single plume-lithosphere interaction
(Rooney, 2017).

An elevated topography is associated with the eruption of the flood basalts in Ethiopia. The flood basalts
are almost exclusively positioned within the elevated regions of the Ethiopian and Somalian plateaus and
the Sarawat Mountains in southwest Yemen (Fig. 1). The Ddynamic topography component supports up
to 1 km of present-day elevation of the Ethiopian and Somalian plateaus, supporting the significant
contribution of mantle convection to the regional uplift (Gvirtzman et al., 2016). Although the uplift
chronology is not easily resolved, recent studies infer it is a long-term feature already present active before
the emplacement of the flood basalts (Sembroni et al., 2016; Faccenna et al., 2019). Regional uplift is
estimated to begin before 40 Ma, with maximal uplifts between 12 and 28 Ma, reaching an average
elevation of 2,500 m (Fig. 4) (Sembroni et al., 2016).

### 4.2. Gulf of Aden

The beginning of continental rifting in the Gulf of Aden relies on the dating of sedimentary sequences,
published in the 90's (see Bosworth et al., 2005 for a review).Onshore outcrops in Yemen (Watchorn et
al., 1998) and in Oman (Roger et al., 1989) and offshore wells (Hughes et al., 1991), suggest that rifting in
the central and eastern Gulf of Aden began at early to mid-Oligocene, within the Rupelian (33.9 - 27.8 Ma).
Syn-rift sediments from the central Yemeni margins indicate that rift flank uplift occurred before any
significant regional extension. The continental rifting climax of continental rifting is estimated between 20
and 18 Ma (Watchorn et al., 1998). Radiometric dating indicates that the margins became stable already
in the Early Miocene (Bosworth et al., 2005), and rift-to-drift transition is interpreted to occur between
~21.1 and ~17.4 Ma (Watchorn et al., 1998). The seafloor spreading center in the Gulf of Aden is
recognizeddeveloped along most of its length and is connected to the mid-ocean ridge in the Indian Ocean
through the Sheba Ridge (Gillard et al., 2021). In the central Gulf of Aden, magnetic isochrons suggest

opening rates of ~27 mm/a prior to 11 Ma, and a slowdown after 11 Ma (Fig. 4). Chron 5C (purple stripes
in Fig. 3; 16.0 Ma) is present along the Gulf of Aden up to the Shukra al Sheik discontinuity (Fig. 3; Fournier
et al., 2010). This implies that the spreading center developed very rapidly, spreading over more than 700
km in less than 1.5 Ma. This fast propagation ceased at the Shukra al Sheik discontinuity. The youngest
magnetic isochrons (2A, 2.6 Ma) are recognized up to ~~longitude~~ 43.9°E in the eastern Gulf of Tadjoura,
~150 km west to the Shukra al Sheik discontinuity, indicating that along this segment, the ridge propagated
westward at an average rate of ~11 mm/a, in the last 16 Ma. Within the Gulf of Tadjoura, no direct
evidence of oceanic spreading was reported in the literature~~to our best knowledge~~.

### 4.3. Red Sea

Sedimentary sequences from offshore drillings suggest that rifting in the Red Sea postdated the rifting in
the Gulf of Aden by a few million years (Bosworth et al., 2005). Independent studies suggest that rifting
had begun simultaneously along the entire Red Sea at late Oligocene-Early Miocene, at ~23 Ma (Plaziat et
al., 1998; Szymanski et al., 2016; Stockli and Bosworth, 2018; Morag et al., 2019). Magnetic isochrons
associated with seafloor spreading are recognized at the southern parts of the Red Sea (Fig 3 ; Girdler and
Styles, 1974). However, oceanic lithosphere is probably abundant along most of the basin (Augustin et al.,
2021). Chron 3 (4.2 Ma) is only present between ~~latitudes~~ 16°N and 18°N, while chrons 2A (2.6 Ma) and 2
(1.8 Ma) are present up to ~~latitude~~ 22°N (Schettino et al., 2016). The recognition of Chron 5 (10 Ma) in the
central Red Sea was recently suggested to mark the beginning of seafloor spreading (Okwokwo et al.,
2022). Structural reconstructions, geodetic measurements, and magnetic anomalies suggest an opening
rate of ~11 mm/a up to ~4.6 Ma, an abrupt increase in the opening rate to ~25 mm/a between 4.6 and 1.8
Ma and a decrease to ~14 mm/a (Fig. 4 ; Schettino et al., 2018). The southern edges of the magnetic chrons
suggest that the southern Red Sea ridge propagated 50 km southwards, between 4.2 to2.6 Ma (~30 mm/a).
Since 2.6 Ma, the Red Sea ridge has not propagated southward, probably due to the decrease in angular
velocity of Danakil relative to Arabia (Fig. 3 ; Schettino et al., 2018).

### 4.4. Main Ethiopian Rift

~~The onset~~Age of faulting and volcanism along segments of the Main Ethiopian rift suggest a diachronous
development of the different segments of the Main Ethiopian Rift (e.g. Bonini et al., 2005). However, there
is no agreement regarding the exact timing of events and even the propagation trend of the rift.
Reconstructions based on magnetic anomalies from the Southwest Indian ridge suggest an upper limit for
the Nubia-Somalia separation at ~19 Ma, including large uncertainties regarding the rates and directions
of the relative motion pre-16 Ma (DeMets and Merkouriev, 2016) (Fig. 4). Geochronological data suggest
that volcanism and rifting in East Africa started at the Turkana depression in southern Ethiopia at 50 Ma
(Varet, 2018) and episodically propagated northwards~~north.~~ H~~h~~owever, it is still a matter of debate if
there is a general propagation pattern or if different segments propagated in different directions (see figs
42-44 in Corti, 2009). Nevertheless, radiometric dating of structural features indicates that extension
commenced at ~11 Ma within the northern Main Ethiopian Rift (Wolfenden et al., 2004).
In summary, regional uplift and flood basalt volcanism in Ethiopia preceded the rifting of the Afro-Arabian
rift (e.g., Rooney, 2017). The rift arms developed diachronically~~at different times~~, with the onset of ~~when~~
rifting in the eastern Gulf of Aden occurring ~~started ~~during the late phases of flood basalt volcanism (at
~30 Ma) whereas rifting in the Red Sea (at ~23 Ma) and the Main Ethiopian Rift (at ~19 Ma) ~~started in a~~
~~lag of~~lagged by ~5-7 Ma~~ after flood basalt volcanism~~.

# 5. Data and Methods

We used bathymetry (Gebco compilation) and topography (SRTM 15+) data to identify morphotectonic features. To highlight and map the architecture of the margins and axes of the rifts, we applied the Difference of Gaussians (Fig. 5) method to the topography and the bathymetry grids (Akram et al., 2017). This method allows a fast and accurate edge detection of elevation using active spatial bandpass filtering. We applied luminance coloring to the resulting grid using the open-source image processing software Gimp.org.

To study density-related shallow crustal structures, we used the satellite altimetry-derived vertical gravity gradient (VGG) model of Sandwell et al. (2014), offering 1 arc-min resolution at offshore regions. As higher frequencies are intensified in the spectral power of the VGG, its anomalies are more source-localized and shallow-sensitive than free-air anomalies. To enhance the edges associated with the VGG, we applied a linear 11-colors colormap, further applied transparency to the VGG map, and projected it on a shaded relief (Fig. 6a).

To study deeper crustal structures and eliminate the topography effect, we used Bouguer gravity anomaly (BGA), derived from the XGM2019 gravity model (Zingerle et al., 2020), calculated with a grid step of 0.1 degrees. The XGM2019 is the most updated global gravity model of the International Centre for Global Earth Models (ICGEM) and is provided in terms of spherical harmonics up to 2159 degrees (Ince et al., 2019; Zingerle et al., 2020). In addition, we applied a linear 240-colors colormap to enhance BGA structures, further applied transparency to the BGA map, and projected it on a shaded relief (Fig. 6b).

To better correlate and discriminate crustal structures and rift features, we considered 1913 earthquake locations (Fig. 3) from the International Seismological Centre catalog with minimum magnitudes ~~above~~ > 4 ML, recorded between 1964 and 2019. To better infer recent tectonic and volcanic activity, we further considered the locations of Quaternary onshore volcanoes (Fig. 3), from the Global Volcanism Program (Smithsonian Institution) and Google Earth mapping.

# 6. Results

## 6.1. Rift margins

The most prominent morphological feature of the rift system is the escarpment along its shoulders. The escarpments mark the rift margin as they distinguish between (1) uplifted pre-rift rocks of the Arabo-Nubian shield or trap basalts sequences and (2) Quaternary arid fluvial sediments or young volcanic sequences, although several continental crustal fragments are present within the Afar Triangle. The edge detection analysis of topography and bathymetry data allows us to outline the rift margins (Fig. 5). This method highlights high frequency details. ~~where in Fig 5. s~~Steep gradients are shown in bright colors and moderate gradients in grey colors (~~in~~ Fig 5)~~.~~.

In the Red Sea, the escarpments are generally continuous with an average rift width of 440 ± 20 km (calculated perpendicular to the Red Sea axis in the study area), and a general increase in rift width from north to south (Fig. 5b). We identify two segments that mark an abrupt change in rift orientation and rift width: (1) Below 15.5°N on the African margin and 18°N on the Arabian margin (segment I in Fig. 5), the escarpment deviates from its general parallel to the Red Sea trend, bending towards the Afar region. The escarpment is characterized by seismic activity from that point on the African side, which is also considered

the northern point of the western Afar margins (Zwaan et al., 2020a). (2) Below 12.5°N on the African
margin and 15°N on the Arabian margin (segment II in Fig. 5), we identify another abrupt change, both in
the orientation and the width of the rift. That point on the African margin is the intersection of the
Tendaho-Goba'ad Discontinuity with the Western Afar Margins (Tesfaye et al., 2003). We note that these
changes are noticeable and similar on the African and Arabian sides (Fig. 5a).
In the Gulf of Aden, the escarpments generally follow the trend of the basin (Fig 5). In the western parts,
the escarpments are irregularless straight and less continuous than those of the Red Sea and generally
reflect the sinistral basin structures. This morphology is well explained by oblique rifting along the Gulf of
Aden (Leroy et al., 2013). The average rift width in the study area is 470 ± 45 km (calculated rift-
perpendicular), with a general eastward increase (565 km at 47.5°E and 420 km at 43.2°E; Fig. 5b). We
recognize an abrupt change in rift width along three lines (III-V in Fig. 5), which are associated with fracture
zones. Along the Somalian margin, prominent sinistral offsets are recognized along lines III and V. This
escarpment segment is a morphological continuation of the Tendaho-Goba'ad Discontinuity lineament,
and is also prominent in the VGG map (Fig. 6a).
Although recognizable in the processed topography map, the rift shoulders are gentler less sharp in the
Main Ethiopian Rift (Fig. 5a). They are prominent in the gravity data as they are associated with VGG and
BGA highs (see profile A in Fig. 9). In the Afar region, the margins show a funnel shape (Fig. 5a). The distance
between the Somalian and Ethiopian escarpments is steadily and monotonically increasing from the Main
Ethiopian Rift to the Tendaho-Goba'ad Discontinuity (Fig. 5b), suggesting that this segment is intact and
non-disturbed by the other arms of the rift system.
In summary, the rift margins of the Red Sea and the Gulf of Aden are interrupted within the proximity to
the Afar triangle, whereas the margins of the Main Ethiopian Rift smoothly funnel into the Afar triangle.
*6.2. Rift axes*
Along the Red Sea and the Gulf of Aden basins, the rift axes are distinctively characterized by deep and
sharp bathymetric troughs, VGG lows, BGA highs, and intense seismic activity (Fig. 3 and Fig 6.). However,
with the proximity to the Afar region, the rift axes change their characteristics.
The rift axis along the Red Sea is outlined by a deep and wide axial trough that ends at 14.5°N,
approximately 400 km from the triple junction (Fig. 7a). South of 14.5°N, we find geophysical evidence
that the rift axis is bent westwards, running meeting the onshore at the Bay of Beylul (white dashed line
in Fig. 7b). The VGG signature and the bathymetry display highs along the walls (50 Eotvos) and lows along
the center (Fig. 7b and profile B). A trail of volcanic islands follows this path (Hanish-Zukur Islands; Fig. 3),
and the alignments of volcanic cones and vents on the islands are orthogonal to the trail of the islands
(Mitchell and Bosworth, (in press); Gass et al., 1973).  A general trend of recent magmatic bodies onshore
meets this line at the Bay of Beylul (Fig. 3). However, major fault sets are not observed in the onshore area
of Beylul (Rime et al., 2023). In addition, a best fit GPS-based rigid block model suggests a block boundary
along this path (Viltres et al., 2020), which is also supported by the fact that the rotation of Danakil relative
to Arabia stopped around L~0.3 Ma (following Schettino et al., 2018 and personal communication). In
addition to, the bent axial segment, a typical gravity signature of the Red Sea rift axis, with a 20 mGal
central BGA peak and 60 – 40 Eotvos VGG side peaks, is also recognized along the connection of the Red
Sea with the Gulf of Aden at Bab al Mandab Strait (13.2°N to 12.3°N; Fig. 7 profile CC'). Nevertheless, no
earthquakes, volcanic activity or faulted bathymetry morphology is found observed along this segment,
thus we propose that this segment is not an active rift axis. However, diluted activity is inferred from the
low and oblique velocity of Arabia in this area (Fig. 3).
In the Gulf of Aden, there is also a distinct change in the characteristics of the rift axis, approximately 400
km east to the triple junction region (Fig. 8). East to the Shukra al Sheik discontinuity, the Gulf of Aden is a
>2,000 m deep basin, which steeply deeps close to the shoreline. Along the basin, the axial trough is
fragmented by oblique left-lateral transform faults (Fig. 3). On the other hand, west to the Shukra al Sheik
discontinuity the basin is shallow (~-700 m). In this section of the Gulf of Aden, the ~-1,700 m deep and
~400 km long curved axial trough impales the Afar triangle at the Gulf of Tadjoura (Djibouti) (Fig. 8). This
axial segment has a distinct gravity signature with 75 mGal central BGA peak and 20 – 35 Eotvos VGG side
peaks, and is characterized by the most intensive seismic activity, perhaps the most intensive in the rift
system, with over 1,000 recorded events with magnitudes > 4 ML (ISC catalog ; Ruch et al., 2021).
In the Main Ethiopian Rift, there are no abrupt changes in the morphology and trend of the rift valley in
the proximity to the Afar triangle (Fig. 9). Instead, the rift valley goes through an elevated dome peaking
approximately 400 km from the triple junction (Fig. 9a). The along-strike profile (profile B in Fig. 9) shows
that the rift valley reaches elevations of more than≤ 2,000 m and is associated with a BGA low of -220
mGal.
In the Afar triangle, the morphology and VGG data indicate two distinguished regions of axial segments
(Fig. 10). (1) Southwest of the Tendaho-Goba'ad Discontinuity, a NE trending valley follows the NE trend
of the Main Ethiopian Rift, characterized by distinct central volcanoes along the axial depression (Fig. 3
and Fig. 10a). (2) Northeast of the Tendaho-Goba'ad Discontinuity, axial segments are composed of NW
trending short segments along volcanic ranges, parallel to the trend of the Red Sea
In summary, the rift axes of the Red Sea and the Gulf Aden drastically change their trend and morphological
characteristics ~400 km from the triple junction. In contrast, the trend and morphological characteristics
of the Main Ethiopian Rift are consistent from the Ethiopian highs up to the triple junction point in Afar.

# 344 7. Discussion

### 345 7.1. The architecture of the intersection region

The Afar triangle is the intersection region of three rift arms: the Gulf of Aden, the Red Sea, and the Main
Ethiopian Rift. Far from the intersection region, the architecture of the rifts, with rift margins parallel to
rift axes, suggest that rigid plate tectonics of the Nubian, Arabian, and Somalian plates controlled their
structural development (Garfunkel and Beyth, 2006; Reilinger et al., 2006; Reilinger and McClusky, 2011;
Schettino et al., 2018). However, the architecture of the intersection region is not simply resolved by rigid
plate kinematics (Garfunkel and Beyth, 2006). Our analysis points abrupt changes of the architectures of
the Gulf of Aden and of the Red Sea rifts, ~400 km from the triple junction. Here, the margins deviate from
their general orientation and show peaks in rift width (segments I to V in Fig. 5) and are not parallel to the
rift axes. The axes themselves deflect from their usual rift-parallel orientation and are curved towards the
direction of the triple junction as they meet the shoreline, forming bays (Fig 7 and Fig. 8). Within the Afar
triangle, northeast of the Tendaho-Goba'ad discontinuity, the margins are fragmented, and there are
multiple, short, and sub-parallel axial segments (Fig. 10).
Fig. 11 shows the mapped rift margins and axial segments. In this study, the term "mapped axial segments"
is not simply correlated with rift axes, especially in the onshore regions. The geology in this regions is quite
complex, including several fault and transfer zones, and, exposing pre-rift rock sequences (e.g., Varet,
2018), however, the mapped axial segments are somewhat correlative with rift axes that had been
suggested based on field observations (e.g., Rime et al., 2023).
Within the Afar triangle, southwest to the Tendaho-Goba'ad discontinuity, the rift margins are continuous
and smooth, and the axial volcanic range generally continues the trend of the axial valley of the Main
Ethiopian Rift, reflecting a sub-perpendicular extension in accordance with the Nubia –Somalia kinematics,
and thus, could be regarded as a rigid plate boundary.
Northeast of the Tendaho-Goba'ad discontinuity, axial segments are generally sub-parallel to the Red Sea
axis (Zwaan et al., 2020b), which led to the interpretation of this region as ~~authors to suggest that this~~
~~region reflects~~ an evolving discontinuity of the oceanic spreading center in the Red Sea (e.g. Tazieff et al.,
1972; Bosworth et al., 2005). Although several focal solutions indicated dextral strike-slip motions in this
area, we don't find other evidence for a typical first-order transform connection between the ridge in the
Red Sea and the continuation of the northern Afar axial segments, offshore Gulf of Zula. Magnetic
isochrons in the Red Sea are mapped over 100 km south of the Gulf of Zula (Fig. 12), and the volcanic ridge
in the southern Red Sea is very active (Eyles et al., 2018). Although earthquake clusters at 16.5°N indicate
strike-slip solutions, supporting a structural connection to the Red Sea axis, these are abundant throughout
the study area (Hofstetter and Beyth, 2003). Alternatively, the jump between the Red Sea ridge and the
axial segments in northeastern Afar could be interpreted as a non-transform discontinuity, however,
second-order discontinuities are usually characterized by <30 km offsets, and here the jump is ~200 km
(Macdonald et al., 1984; Carbotte et al., 2016). Thus, there is no structural evidence to relate the axial
volcanism in the Afar triangle to the Red Sea spreading center. This conclusion agrees with the study of
Rime et al. (2023), which suggests a northward propagation of the rift in the Danakil Depression mainly
supported by younging ~~trend~~ of magmatic products, and rifting ages ~~and other arguments~~.

Commented [RI1]: According to Wiktionary:
**younging** (uncountable):
1. (*geology*) The direction in which stratigraphy becomes younger, for a particular formation

The architecture of the intersection region northeast to the Tendaho-Goba'ad discontinuity reflects a
rugged connection of the Red Sea and the Gulf of Aden arms to the Main Ethiopian Rift and is characterized
by diffuse deformation rather than sharp plate boundaries. A recent model based on GPS observations
(Viltres et al., 2020) reveals a diffuse Danakil - Nubia boundary with inter-rifting deformation over > 100
km wide zone. The Danakil microplate extends to the Hanish-Zukur Islands at its southern edge (∼13.8°N)
with no precise/sharp boundary (Fig. 3). The Danakil microplate is rotating counterclockwise (at a mean
rate of 1.5° ± 0.6°/Ma for the last ~7 Ma ; Manighetti et al., 2001), while the Ali-Sabieh block, south of the
Gulf of Tadjoura, is rotating clockwise (15° between 8 to 4 Ma ; Audin et al., 2004), described as a "saloon-
doors" mode of opening (Fig. 11; Kidane, 2016).
The concept of segments of localized strain, which are spread over a broad zone in Afar was noted from
many indicators including diking events, structural geology, seismology and geodesy (Keir et al., 2011; Pagli
et al., 2014, 2018; Doubre et al., 2017). Analogue models demonstrated that the plate interactions in Afar
results in a broad zone of localized extension (Maestrelli et al., 2022).
Hence, the architecture of the intersection region of the rift arms discloses a ~150,000 km$^2$ complex region,
in which diffuse boundaries and microplate rotations link the three rift arms (Fig. 11). Accordingly, a
genuinely single triple junction point, in the sense of a three-rift arms intersection point, cannot be
specified for this system, and multiple triple junctions could be considered  (e.g., see tectonic models in
Viltres et al., 2020). The difficulty of defining sharp plate boundaries within Afar was discussed in many
works (e.g., Barrberi and Varet, 1977 and references therein). Nevertheless, we agree that the intersection
point of the Ethiopian rift valley and the Tendaho-Goba'ad Discontinuity could be regarded as the 'main'
junction point of the rift system, as the deformation characteristics between the northern Main Ethiopian
Rift and the diffuse zone on the Gulf of Aden – Red Sea rifts are most distinctively changed there (Tesfaye
et al., 2003).

### 7.2. Spatial constraints in the development of the plume-rift system

The mapping of the rift margins and axial segments allows us to draw two spatial constraints in the
development of the plume-rift system:
(1) The first is the connection of the Main Ethiopian Rift to the Gulf of Aden - Red Sea rifts by a
northeastward propagation. Since the divergence between Nubia and Somalia is sub-perpendicular to the
strike of the northern Main Ethiopian Rift, its propagation direction is not entirely dictated by the
kinematics (Tesfaye et al., 2003; Wolfenden et al., 2004; Bonini et al., 2005; Keranen and Klemperer, 2008;
Abebe et al., 2010). The margins of southeast Afar show symmetric, continuous, and smooth curved
trends, from the elevated regions of the Main Ethiopian Rift to the Tendaho-Goba'ad Discontinuity (Fig.
5). With respect to the northeastward trend of the Main Ethiopian rift, the Somalian margin is curved
clockwise, following the Ali-Sabieh sense of rotation (Kidane, 2016), whereas, the Ethiopian margin is
curved counterclockwise, like the Danakil sense of rotation (Fig. 11; Schult, 1974). This architecture could
be understood in terms of fracture mechanics by the reorientation of a propagating fracture near a pre-
existing fracture. Strain analysis indicates that a propagating fracture would curve parallel to the pre-
existing fracture under a tensional stress field due to free surface boundary conditions induced by the
open pre-existing fracture (Dyer, 1988). In analogy, the architecture of the study area express a smooth
linkage of the Main Ethiopian Rift to the pre-existing Gulf of Aden-Red Sea rifts by a northeastward
propagation. Hence, this implies that a triple junction formed at a late stage, when all three arms were
already significantly developed. This conclusion agrees with structural geochronology within the northern
Main Ethiopian Rift, showing that extension in the northern Main Ethiopian rift commenced at 11 Ma
(Wolfenden et al., 2004).
(2) The second spatial constraint is the abandonment of an early tectonic connection between the Red Sea
and the Gulf of Aden through the Bab al-Mandab Strait. As the VGG and neovolcanic activity indicate that
the Red Sea axis currently enters Afar at the Bay of Beylul (see section 6.2), we find arguments for an
earlier tectonic connection between the Red Sea and the Gulf of Aden through Bab al-Mandab Strait: (i)
South of 13.2°N and up to the connection to the Gulf of Aden (12.3°N), BGA and VGG depict the typical
and previously defined gravity signature of the rift axis (Fig. 7 and Fig. 8; see section 6.2). (ii) The submarine
channel north to the Hanish Island (Fig 7, 13.4°N) shows no association with modern water currents and
possibly formed by faults in the subsurface (Mitchell and Sofianos, 2018). (iii) This is the straight
continuation of the Red Sea axis, along which the basins are curtly connected (Fig. 1). Thus, it is reasonable
proposing that it was the tectonic connection ~~in~~ durring the early stages of rift development. Likewise,
reconstructions suggest that the Danakil microplate started to rotate in the Middle Miocene (~10 Ma),
when Arabia was already separated from Africa (Collet et al., 2000; Schettino et al., 2016; Rime et al.,
2023). Those reconstructions show that the pre-Middle Miocene divergence was focused along Danakil
and Arabia at the southernmost Red Sea. This suggests that the present deflection of the rift axes at the
tip of the Gulf of Aden and the Red Sea marks a tectonic reorganization in this region.
Adopting the fracture propagation analog postulated here for the northeastward propagation of the Main
Ethiopian Rift, implies that the new stress conditions in Afar may be responsible for the abandonment of
the tectonic connection between the Red Sea and the Gulf of Aden. Rime et al. (2023) suggested that the
deposition of lacustrine sediments (Chorora Fm) recorded marks the development of the Main Ethiopian
Rift in Afar. They point out that these sediments were deposited coeval with the individualization of the
Danakil Block, and thus to the decrease of the extensional tectonic activity at the southernmost Red Sea
rift.
These two spatial constraints suggest that the onset of the triple junction occurred at a late stage when
the three rift arms were already developed and the Red Sea was tectonically connected to the Gulf of
Aden, ~250 km away from the present-day triple junction (Fig. 13). The onset of the triple junction marked
a tectonic reorganization and microplate formation. As a result, the Gulf of Aden and the Red Sea arms are
not smoothly connected to the Main Ethiopian Rift, and a vast area of diffuse and complex deformation
developed within the intersection region.

### 7.3. Mechanisms for plume-rift association

The temporal constraints regarding the development of the plume-rift features, summarized in section 4,
together with the two spatial constraints inferred in this study, allow us to examine the causal relationship
between the activity of the Afar plume and rifting. Our insights suggest that neither 'active' nor 'passive'
rifting mechanisms are solely consistent with observations. Passive rifting models fail to explain the plume-
rift association mainly because the flood basalt volcanism cannot be attributed to a passively rising
asthenospheric mantle beneath a stretched and thinned lithosphere, as dynamic uplift in Ethiopia is a long-
lasting process that preceded flood basalt volcanism (Sembroni et al., 2016). Hence, rifting and associated
subsidence are subsequent to flood basalt volcanism (Fig. 4). The estimations that the Ethiopian plateau
was elevated ~1 km before flood basalts (Fig. 4) coincide with active plume-head predictions (Campbell
and Griffiths, 1990). Moreover, the passive model does not explain why a triple junction is located within
the flood basalts area, as rifting in the Red Sea and Gulf of Aden are at an oblique angle to the former
sutures (Buiter and Torsvik, 2014).
On the other hand, active models are not in line with the progressive development of the rifts, mainly
because the flood basalts region cannot be considered a center or a nucleus, from which rift arms spread,
as expected in an actively generated triple junction. Numerous studies noted that the tectonic
development of the Afar region is not compatible with a simplified model of rift arms that simultaneously
spread away from a triple junction (see Section 5.2 in Rime et al., 2023 for a review). The triple junction
was the last feature to develop in the system, by the propagation of the Main Ethiopian Rift towards Afar,
followed by a tectonic reorganization including the abandonment of a former tectonic connection
between the Red Sea and the Gulf of Aden. By this time, the rift arms had already developed, and the
break-up between Africa and Arabia had already been accomplished between Africa and Arabia. This
tectonic reorganization cannot be attributed to the development of gravitational forces exerted by the
plume head (Hill, 1991), as it occurred ~20 Ma after flood basalts magmatism. That rules out the possibility
that the arrival of the Afar plume directly led to the formation of the triple junction and the rift arms did
not spread from the plume region.
We propose a scenario in which rifting was triggered by a plume-induced plate rotation (Fig. 2c). Numerical
simulations suggest that horizontal asthenospheric flows due to the arrival of a plume head at the base of
the lithosphere induce a plume-push force that can accelerate plates by several cm yr$^{-1}$ (van Hinsbergen
et al., 2011, 2021; Pusok and Stegman, 2020). In this scenario, flood basalt volcanism would be
synchronous to an abrupt plate speed-up and thus to new remote stress conditions. In the case of the
Indian plate, at least two episodes of massive flood basalt volcanism, Morondava LIP (~94 Ma) and Deccan
traps (67 Ma), are associated with plume-derived plate acceleration, and a drastic change in the tectonic
framework (van Hinsbergen et al., 2011, 2021; Cande and Stegman, 2011; Pusok and Stegman, 2020).
Further, torque balance modeling simulating the horizontal forces generated from a point source (plume
head) suggests that horizontal plume-push can force a significant plate rotation and, consequently, initiate
new plate boundaries (van Hinsbergen et al., 2021).
In the Afro-Arabian rift, indeed new plate boundaries formed after the arrival of the large Afar plume and
a significant plate rotation of Arabia around a nearby pole characterizes the Arabian continent (Joffe and
Garfunkel, 1987; Viltres et al., 2022). Magnetic anomalies and structural reconstructions suggest that the
rotation around a nearby pole already characterized Arabia since the Oligocene (Fournier et al., 2010;
Schettino et al., 2018). Additionally, the beginning of intensive volcanism in the north-western Arabian
plate (Harrat Ash Shaam) at Late Oligocene (Ilani et al., 2001), reflected a change in mantle-crust
interaction and intracontinental extension within the Arabian plate, adjacent to the arrival of Afar plume
(Garfunkel, 1989). In the Harrat Ash Shaam volcanic field, diking directions from Miocene to recent ages
record the rotation of Arabia (Giannerini et al., 1988), suggesting that already during the first stages of
volcanism the Arabian plate was rotating around a nearby pole.
The arrival of the Afar plume was also accompanied by a slowdown of Africa (Le Pichon and Gaulier, 1988).
By this time, Africa collided with Eurasia in the west, explaining its slowdown (Jolivet and Faccenna, 2000)
and increased intraplate volcanism (Burke, 1996). However, this collision of Africa and Eurasia cannot
simply resolve the change in the rotation of Arabia as the Arabian continent collided with Eurasia not
earlier than ~18 Ma (Su and Zhou, 2020), although some authors suggested that asymmetrical along-
trench entrance of continental material could lead to an intraplate extension similar to those that
generated the Africa-Arabia break-up (Bellahsen et al., 2003). Faccenna et al. (2013) ~~already~~ showed that
plume-push from the Afar area resolves the present-day plate kinematics in the Middle East, particularly
the anti-clockwise toroidal pattern of the Arabia–Anatolia–Aegean system. The importance of active
upwelling in Afar to lateral mantle flow below Arabia is also illustrated by shear-wave splitting, indicating
a general N-S anisotropy in the mantle (Qaysi et al., 2018). Stamps et al. (2014) calculate~~d~~ the current
driving forces for the Nubia-Somalia divergence and found that gravitational potential energy is the most
significant force, stronger by an order of magnitude than forces from basal shear tractions of mantle
convection. They point out that the gravitational potential energy is sufficient to sustain present-day rifting
in East Africa but not to initiate rupture of continental lithosphere. In the case of the Arabian plate, basal
shear tractions are expected to be higher due to the orientation of northward-directed mantle flow
(Faccenna et al., 2013).
Plume-induced plate rotation settles the facts that regional uplift and flood basalt volcanism shortly
preceded rifting (Sembroni et al., 2016) together with the insight that rifting was developed by far field
forces and plate kinematics (Autin et al., 2013; Bosworth and Stockli, 2016). It also explains why the rifts
intersect within the plume region as the lithosphere in this region was weakened by the hot plume material
(François et al., 2018). Finally, it explains the delayed development of the Main Ethiopian Rift and the late
onset of the Afar triple junction by its northwestward propagation, as these were controlled by the slower
kinematics of the Somalian plate rather than dynamic forces. In this tectonic scenario~~manner~~, 'active' and
'passive' mechanisms are coupled and have positive feedback, allowing a close occurrence of flood basalt
volcanism and continental break-up, alongside passive rifting.

# 8. Summary and Conclusions

We reviewed the geologic setting of the Afro-Arabian rift, in which vast regions of flood basalts and ongoing continental break-up are superimposed, aiming to infer a causal relationship between the activity of the deep-seated Afar plume and crustal break-up. We explored the R-R-R triple junction between the Gulf of Aden, the Red Sea, and the Main Ethiopian Rift that divides the large Cenozoic plume-related flood basalt series in Ethiopia and Yemen. Based on ~~a synthesis and~~the interpretation of topography, bathymetry~~,~~ and gravity, and the integration of magnetic anomalies, earthquakes, and volcano distribution, we mapped the margins and axes of the rift arms.

Our results show that the terminations of the Gulf of Aden and the Red Sea arms are rough and irregular, in contrast to the symmetric, continuous, and smooth architecture of the Main Ethiopian Rift. The triple junction formed by the northeastward propagation of the Main Ethiopian Rift and the abandonment of the tectonic connection between the Red Sea and the Gulf of Aden through Bab al-Mandab Strait. This suggest a progressive development of a broad region of diffuse deformation at the intersection area. The onset of the triple junction was the last feature to develop in the plume-rift system after all rift arms were sufficiently evolved and the break-up between Africa and Arabia was already accomplished.

This progressive development does not align with the classic active rifting model, which predicts a plume-generated triple junction at the locus of the rift development~~, from which the rifts develop~~. Nevertheless, the classic passive rifting model fails to explain the chronological evidence, as flood basalts probably erupted on elevated topography before rifting started. We discuss a scenario of plume-induced plate rotation in which the arrival of the Afar plume triggered the rotation of Arabia around a nearby pole that characterizes the system since the Oligocene. We argue that plume-induced plate rotation better explains the progressive development of the plume-rift system in the Afro-Arabian rift.

# 9. Data availability

The bathymetry and topography data used in this study was retrieved from GEBCO Compilation Group (2021), available at https://www.gebco.net/data_and_products/gridded_bathymetry_data/#area.

The VGG data used in this study is available at https://topex.ucsd.edu/grav_outreach/.

The BGA data used in this study is available at http://icgem.gfz-potsdam.de/calcgrid; model XGM2019e-2159, 'gravity_anomaly_bg'.

Earthquake data was retrieved from the International Seismological Centre (2020), On-line Bulletin, https://doi.org/10.31905/D808B830.

Quaternary onshore volcano locations were retrieved from the Global Volcanism Program, Smithsonian Institution, available at https://volcano.si.edu/volcanolist_holocene.cfm.

Magnetic anomalies data is available at
https://figshare.com/articles/dataset/Transcurrent_Regimes_During_Rotational_Rifting_New_Insights_from_Magnetic_Anomalies_in_the_Red_Sea/14743272.

# 10. Author contribution

RI carried out the study and wrote and revised the original draft of this paper. PH and NA provided conceptual assistance, helped in writing and reviewed the manuscript. JE mentored the study, took care of administration, and reviewed the manuscript.

# 11. Competing interests

The contact author has declared that neither of the authors has any competing interests.

# 12. Acknowledgments

This work was supported by the grants from Minerva Fellowship to R. I. We thank Neil Mitchell and Valentin Rime for their helpful discussion throughout the open discussion process. We wish to thank Antonio Schettino and Derek Keir for their review which helped improving the manuscript. We thank the editors of Solid Earth for helpful comments and review process.

# 13. Figure captions

**Fig. 1.** Elevation map of the study area, showing the general plate tectonic configuration (from USGS and from Viltres et al. (2020) in the Afar region) and Cenozoic volcanics (modified from Varet, 1978; Davison et al., 1994; Beyene and Abdelsalam, 2005; Bosworth and Stockli, 2016) Black arrows indicate GPS velocities in respect to Nubia (modified from Reilinger et al., 2006).

**Fig. 2.** Schematic mechanisms for plume-rift association in the Afro-Arabian rift. (a) Active mechanism (e.g., Campbell and Griffiths, 1990). The plume head impinge and erode the base of the lithosphere, which prompt uplift and decompression melting. These introduce internal extensional forces at the crust, leading to break-up.. (b) Passive mechanism (e.g., White and McKenzie, 1989). Rifting is initiated solely by the remote stresses, regardless of the underlying plume. In this mechanism, the production of massive volcanism is allowed when the thinned and stretched lithosphere is underlaid by the thermal anomaly in the mantle. Flood basalts volcanism is generated by decompression melting of the passively rising hot asthenospheric mantle. (c) Plume-induced plate rotation (van Hinsbergen et al., 2021). Plume push forces sourced by the drag of the flowing asthenosphere add up to the remote stresses to change the plate kinematics. In this mechanism flood basalts volcanism is actively controlled, however, rifting is triggered by the new plate kinematics.

**Fig. 3.** Map of the Afar region showing magnetic isochrons (modified from Fournier et al., 2010; Bridges et al., 2012; Schettino et al., 2016), earthquake locations (from ISC catalog), Holocene onshore volcano locations (from GVP catalog and Viltres et al. (2020)) and recent volcanism (modified from Keir et al., 2013).

**Fig. 4.** Elevation of the Ethiopian–Yemen plateau (grey boxes, after Sembroni et al., 2016; Faccenna et al., 2019), volcanic episodes (orange and red bars) and opening rates of the rift arms (blue lines, modified from Fournier et al., 2010; DeMets and Merkouriev, 2016; Schettino et al., 2018). Dashed lines indicate estimations from geological observations and solid lines from magnetic anomalies.

**Fig. 5.** (a) Difference of Gaussians applied to topography and bathymetry showing rift margins (black lines). White dashed lines indicate peaks in rift width. TGD is the Tendaho-Goba'ad Discontinuity. SSD is the Shukra al Sheik discontinuity. Black dots indicate earthquake locations (ISC catalog). (b) Rift widths, calculated in rift-perpendicular directions.

**Fig. 6.** Gravity data of the Afar region. (a) Vertical gravity gradient from Sandwell et al. (2014). Bouguer anomaly model from ICGEM, XGM2019e (Zingerle et al., 2020).

**Fig. 7.** Bathymetry (a), vertical gravity gradient (b) and Bouguer anomaly (c) in the southern Red Sea. Black dots indicate earthquake locations (ISC catalog). (d) Profiles across rift axis.

**Fig. 8.** Bathymetry (a), vertical gravity gradient (b) and Bouguer anomaly (c) in the Western Gulf of Aden. Black dots indicate earthquake locations (ISC catalog). (d) Profiles across rift axis.

**Fig. 9.** Topography (a), vertical gravity gradient (b) and Bouguer anomaly (c) in the northern Main Ethiopian Rift. Black dots indicate earthquake locations (ISC catalog). (d) Profiles across (AA') and along (BB') the rift valley.

**Fig. 10.** Topography (a), vertical gravity gradient (b) and Bouguer anomaly (c) in the Afar triangle. Black dots indicate earthquake locations (ISC catalog). TGD is the Tendaho-Goba'ad Discontinuity. (d) Profiles SW (AA') and NE (BB') to the TGD.

**Fig. 11.** Rift margins (solid white lines) and axial segments (long dashed black lines) in the Afar region. Black dots indicate earthquake locations (ISC catalog). TGD is the Tendaho-Goba'ad Discontinuity.

**Fig. 12.** Tilt-angle derivative map of magnetic anomalies, projected on a shaded relief after Issachar et al. (2022). Purple colures represent positive angles and green colors represent negative angles. White dashed lines indicate magnetic stripes (Schettino et al., 2016).

**Fig. 13.** Synthesis of the progressive development of the rift intersections.

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

## 15. Figures

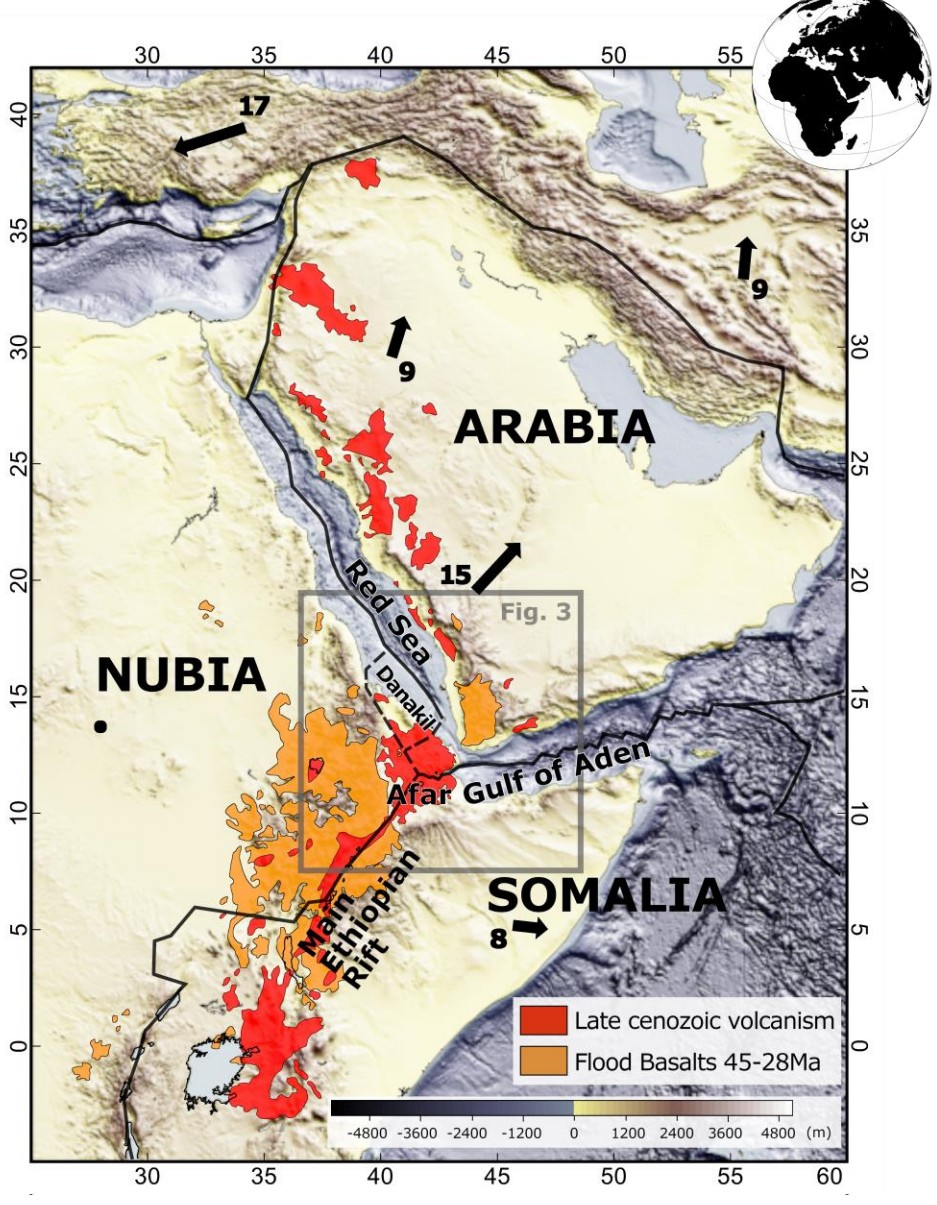

Fig. 1.

Active            Passive            Plume-induced rotation

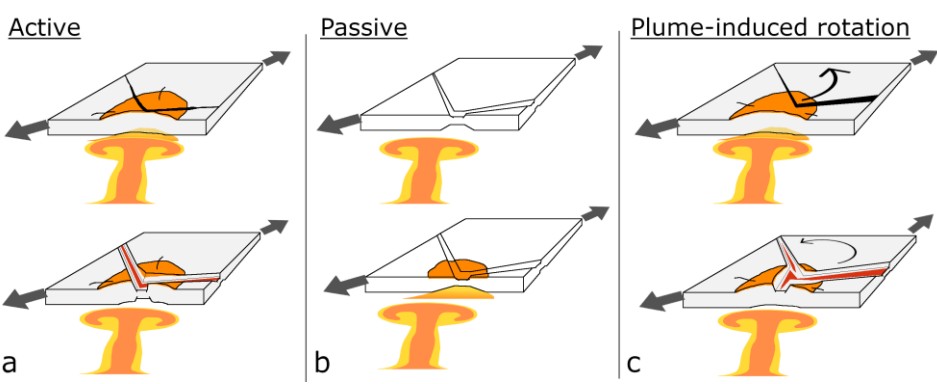

Fig. 2.

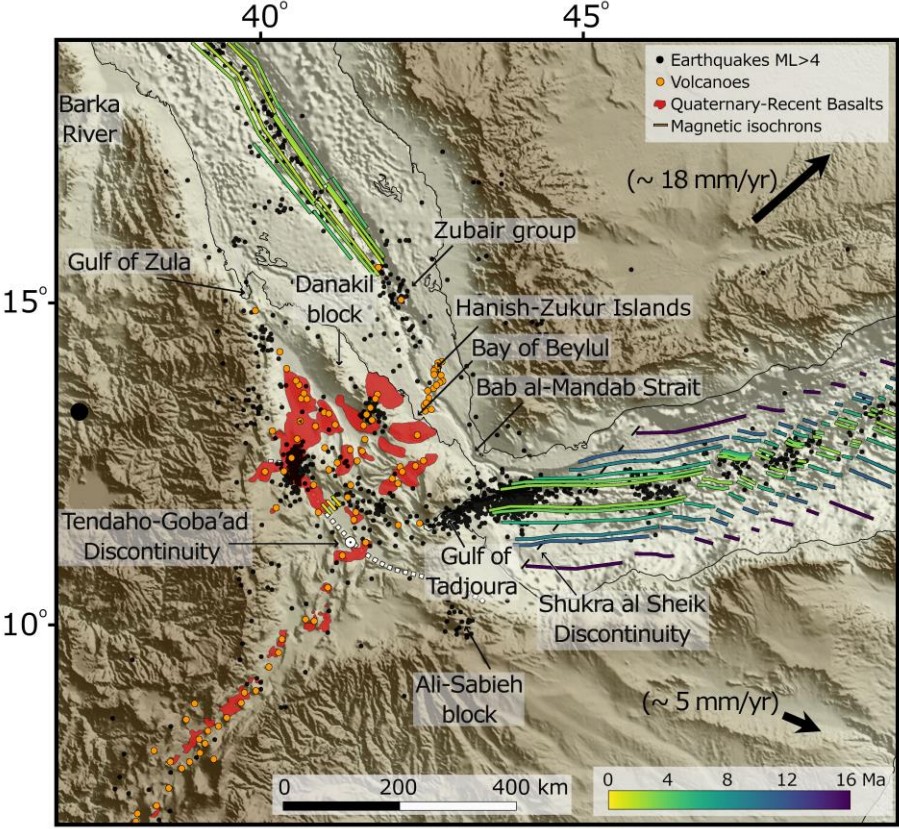

Fig. 3.

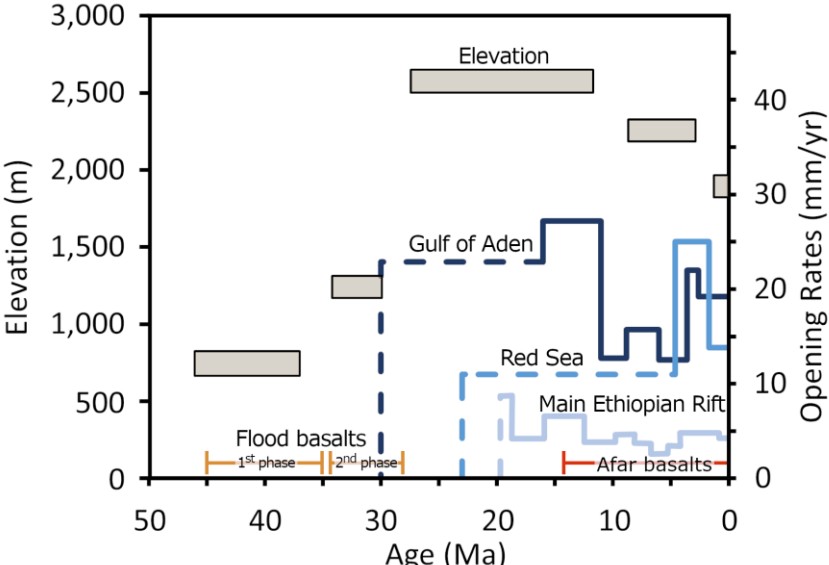

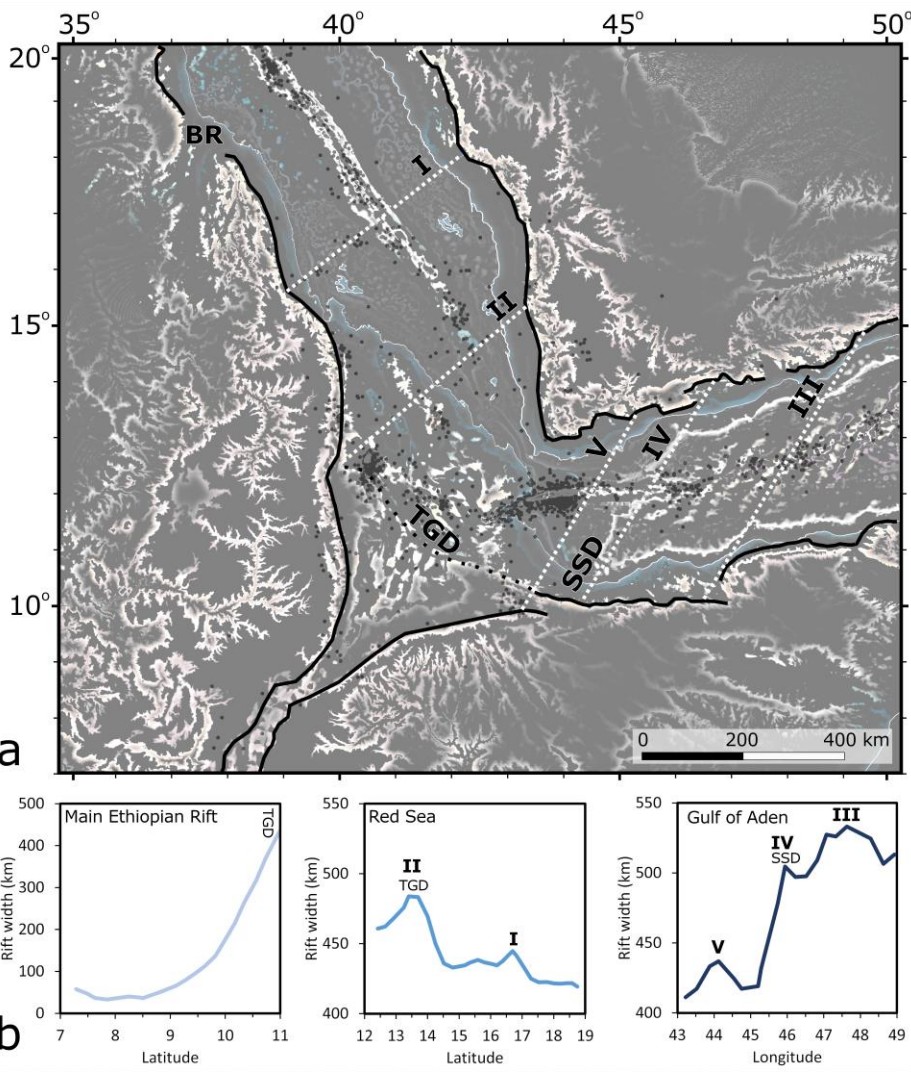

Fug. 5.

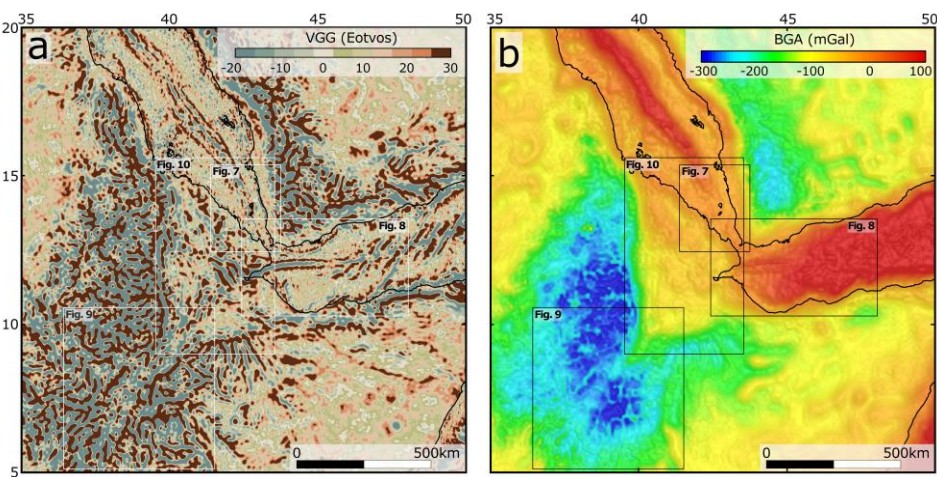

Fig. 6.

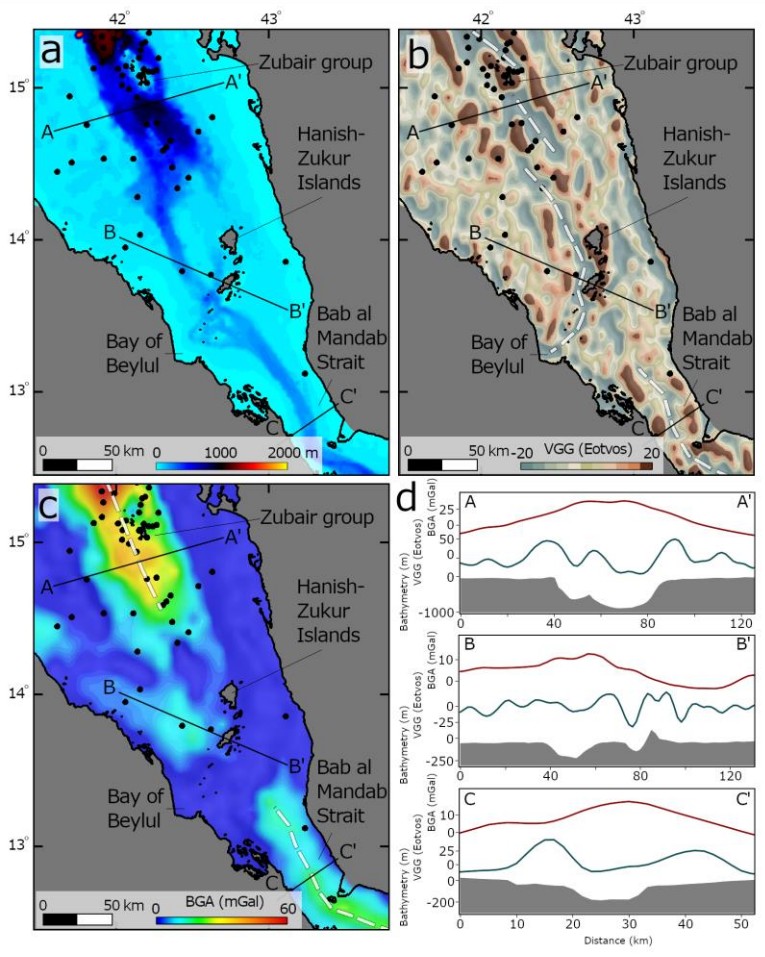

Fig. 7.

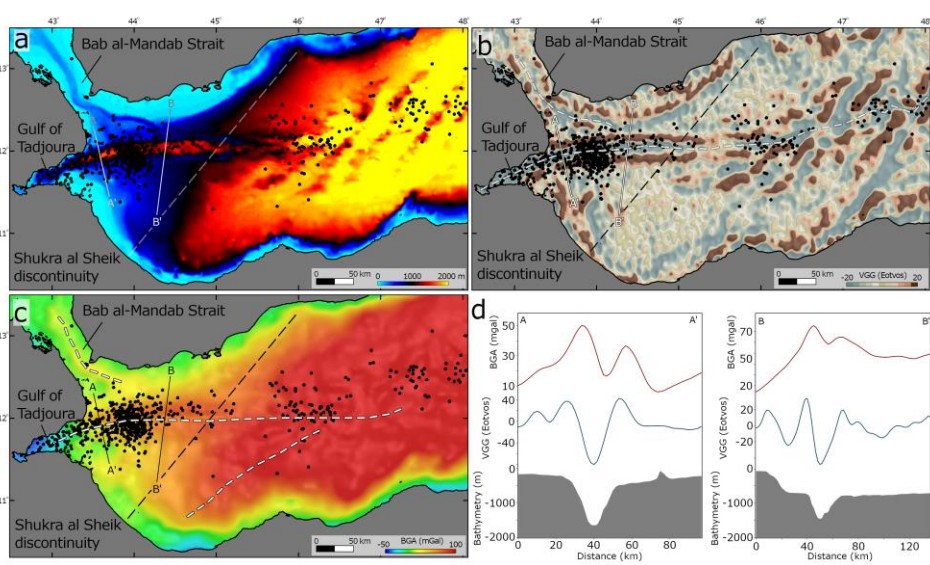

Fig. 8.

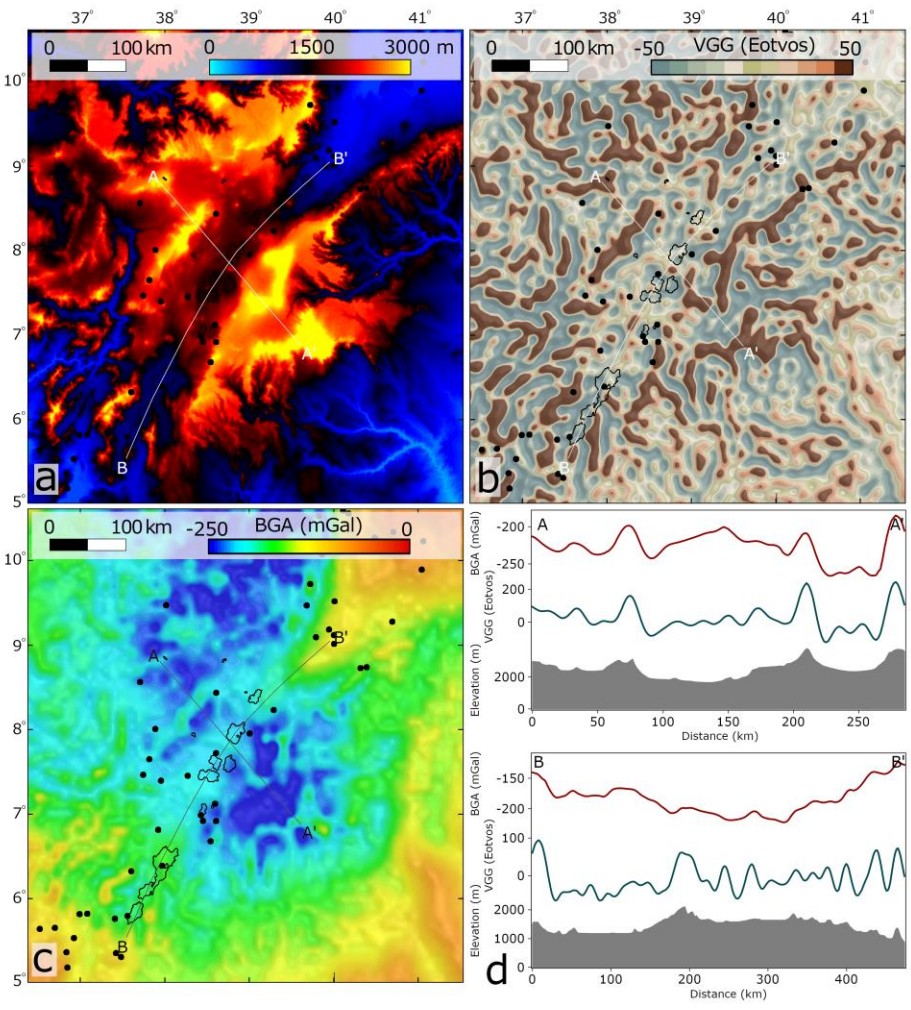


Fig. 9.

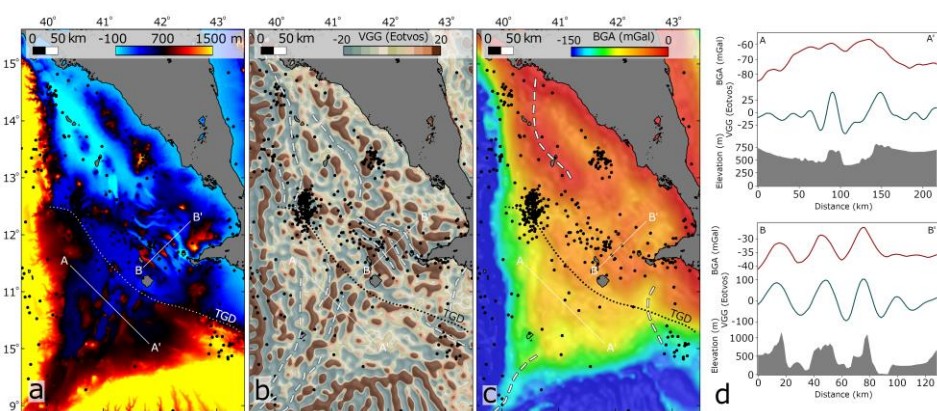


Fig. 10.

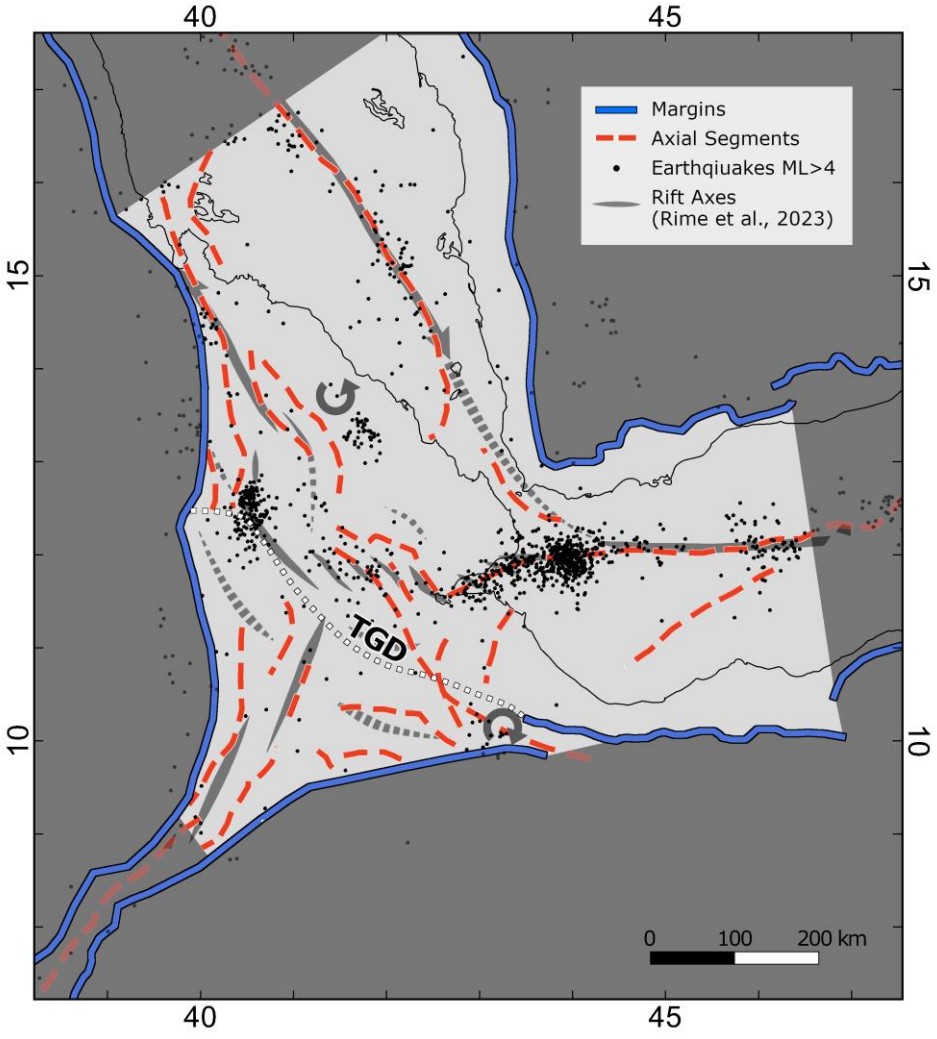

Fig. 11.

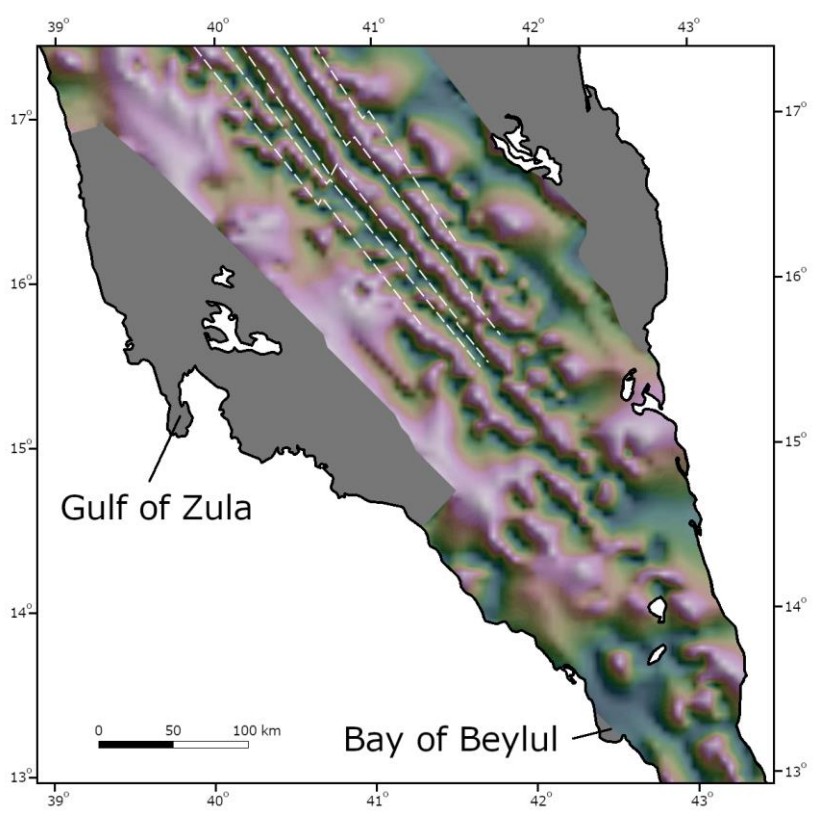

Fig. 12.

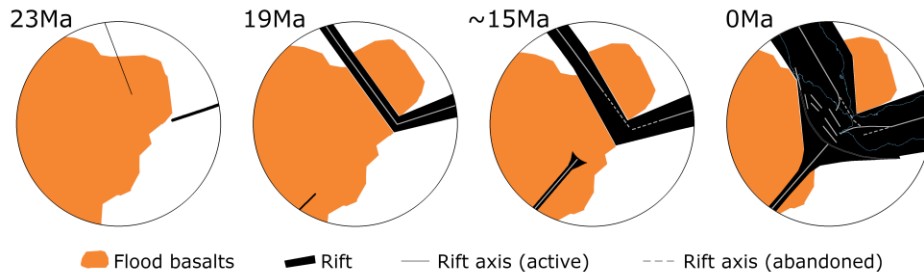

Fig. 13.