# Peer review of "Rift and plume: a discussion on active and passive rifting"

_EGUsphere, 2023_

## Referee Comment (RC2)

Further comments on *Rift and plume: a discussion on active and passive rifting mechanisms in the Afro-Arabian rift based on synthesis of geophysical data*, by Ran Issachar, Peter Haas, Nico Augustin, and Jörg Ebbing

Dear Ran, I appreciate that you are addressing my comments. Below please find some specifications.

1. Regarding the work of Bridges et al. (2012), I think that you have stretched their results writing "Two magnetic isochrons have been recognized in the Tendaho graben, indicating young oceanization in central Afar". In reality Bridges's et al. (2012) observations suggest that transitional crust in Afar has developed a significant amount of remanent magnetization in addition to the induced component. This result does not imply the presence of oceanic crust, it only indicates that the effusive and intrusive rocks of this area have been magnetized in a way that resembles what happens at oceanic spreading centers. Linear magnetic anomalies can form on oceanic crust, on serpentinized outcropping mantle (e.g., southern North Atlantic), and even on continental crust when it is intruded by dykes and characterized by pervasive volcanism.

2. I am confused when you claim that the "rapid propagation was halted in the last 2.6 Ma" or that "southward propagation [of the spreading center] is halted between isochrons 2 and 2A". Below is a plot of the angular velocity of Danakil relative to Arabia since 14 Ma (according to my published calculations and magnetic modelling, of course). As you can see, the velocity decreased linearly between 14 Ma and C2y (1.77 Ma). If you extrapolate this plot to more recent times, you reach the conclusion that the relative motion between Danakil and Arabia did not stop until ~0.3 Ma.

[Figure]

3. Regarding the possibility of a rift jump from the southern Red Sea to the Afar area, this is an intriguing theory, which is supported by the plot shown above. I suggest to find strong geological evidence of a left-lateral strike-slip fault through Danakil to prove your assertion.

Best regards,

A. Schettino

---

## Author Response (AR1)

Dear Editor,

Please see our revised manuscript "Rift and plume: a discussion on active and passive rifting mechanisms in the Afro-Arabian rift based on synthesis of geophysical data".

We carefully considered the comments and suggestions of the reviewers and community, and replied on the public discussion. We feel that the comments and following discussion were very helpful and improved the manuscript significantly.

In addition to our replies in the public discussion, for each comment of the reviewers you can find below our explanation of how we specifically revised the manuscript.

Figures 1, 3 and 11 are modified in the revised manuscript.

Hope you will find the revised manuscript suitable for publication in Solid Earth

On behalf all author,

Ran Issachar

Comment by Antonio Schettino:

R38-40: "Continental flood basalts (...) are associated with extensive volcanism during short time intervals, brought to the surface by deep-seated mantle plumes". I am not sure that any LIP is associated with the presence of a deep mantle plume. The authors should consider at least the valuable contributions by Don Anderson (e.g., Anderson, 1994 ; Anderson, 2005).

- We changed the text accordingly and referred also to the contributions by Don Anderson.

R41-42: "Observations indicate a close temporal and spatial occurrence between the eruption of flood basalts and continental break-up". Are the authors suggesting that any eruption of flood basalts is associated with continental break-up? As pointed out by Buiter & Torsvik (2014), this is certainly false.

- We modified the text to avoid confusion.

R42-44: "when reconstructed back to their original plate tectonic configuration, a R-R-R triple junction is found within the flood basalts areas". This assertion is incorrect if considered as a general statement. We have examples of RRR triple junctions in magma poor conditions (e.g., in the southern North Atlantic during the Cretaceous).

- We modified the text to avoid confusion.

R70: "deep mantle convection and its interaction with the Earth's crust". It is more correct to say "Earth's lithosphere".

- We changed the text

R73-76: "This led Morgan (1971) to speculate that deep mantle convection has a significant role in accelerating the overlying tectonic plates. Nevertheless, it was later realized that slab-pull provides the main driving force for plate motion. Furthermore, plumes are thought to have a major role in plate tectonics, triggering rifting by weakening the upper lithosphere". This is somewhat confusing. The first sentence seems to recover and accept Morgan's (1971) proposal that "deep mantle convection has a significant role in accelerating the overlying tectonic plates". The successive sentence recalls the contrasting modern view that the main driving force of plate tectonics is slab pull, but without citing Forsyth & Uyeda (1975) work. Finally, the last sentence, which starts with "Furthermore" but is unrelated to the previous sentences, suggests that plumes have a major role in plate tectonics because they are responsible for the weakening of the upper lithosphere. This seems to imply that in absence of mantle plumes we would have no weakening of the "upper lithosphere" (why "upper"?, does the lower lithosphere remain strong?). In reality, the continental litosphere is weak (and can be extended) because of the presence of water (for a short discussion on this point see Schettino & Ranalli, 2023). When a mantle plume is present, it causes thinning, that is, a rise of the isotherms, not weakening.

- We edited the text and removed the unrelated sentence to avoid confusion.

R127-128: "Six pairs of magnetic stripes are recognized along the Gulf of Aden ridge". It would be more correct to avoid the textbook term "magnetic stripes" and use instead "magnetic anomalies". Furthermore, it should be "the Gulf of Aden", not "the Gulf of Aden ridge".

- We changed the text.

R148-160: "Two magnetic isochrons have been recognized in the Tendaho graben, indicating young oceanization in central Afar (Bridges et al., 2012)". This is really a stretch of Bridges's et al. (2012) thinking. It seems that the authors have mis-interpreted the important results of Bridges's et al. (2012) paper.

- We understand and agree. We modified the text.

R179: "no recent data was published". It should be: "no recent data were published".

- We changed the text.

R209-211: "The southern edges of the magnetic chrons suggest that the ridge rapidly propagated southwards, with rates of ~30 mm/yr, between chrons 3 (4.2 Ma) and 2A (2.6 Ma). However, the rapid propagation was halted in the last 2.6 Ma". I don't understand this observation. The Red Sea ridge is composed by two independent branches. The southern ridge separates Arabia from Danakil with a rotation pole that is located in the Gulf of Aden. Consequently, the linear velocity increases northwards and the ridge propagated southwards, as correctly stated by the authors. However, the northern ridge separated Arabia from Nubia about a pole that is located ~50 km south of El Alamein. Therefore, in this case the linear velocity increases southwards and the ridge propagated northwards. Incidentally, I also note that this manuscript does not attribute much importance to the role of Danakil.

- We understand the point and agree. We changed the text.

R289-290: "South of latitude 14.5°, we find geophysical evidence that the rift axis is bent, entering the Afar region at the Bay of Beylul (latitude 13.3°)". This is a very questionable interpretation. In my opinion, although Fig. 7a and 7b show the presence of two transform segments (in SW-NE direction), the rift axis does not enter the Danakil microplate and continues in SSE direction. The proposed boundary of the Danakil plate shoud be justified by new fieldwork, because simple visual analysis of geophysical maps could not be convincing for many readers.

- We added new geological evidence of volcanic cones and vents orientations in the Hanish islands to support the geophysical interpretation.

R297-298: "Nevertheless, this segment is not an active rift axis as no earthquake, volcanic or bathymetrical expression is associated with it (Fig. 3)". This is not an argument, as the velocity of separation between Arabia and Danakil is very small in this area (less than 10 mm/yr) and very oblique with respect to the axis. I tested the possibility of a NE–SW strike–slip structure that transfers extension from the Red Sea ridge to Afar through Danakil but any kinematic test failed (this is discussed in Schettino et al, 2016).

- We modified the text and added points raised by the reviewer in the discussion.

R332-334: "the architecture of the intersection region northeast to the Tendaho-Goba'ad discontinuity is more complex and is not simply resolved by rigid plate kinematics". Clearly, a region characterized by stretching and rifting cannot be described in terms of rigid rotations. People involved in plate kinematics studies use rigid rotations to describe the motion of plate interiors, not of deforming margins.

- We agree, but this is a very broad region. This citation is from Garfunkel and Beyth (2006).

R340-342: "Axial segments are generally sub-parallel to the Red Sea axis and not to the rift margins, which led authors to suggest that this region reflects an evolving discontinuity of the oceanic spreading center in the Red Sea". I don't understand this sentence.

- We rephrased the text.

R342-344: "we don't find any evidence for a transform connection between the ridge in the Red Sea and the continuation of the northern Afar axial segments, offshore Gulf of Zula". This sentence is also

confusing. E-W dextral strike-slip faults in the area north of the Gulf of Zula are documented by several CMT fault plane solutions.

- We edited the text.

R396-397: "reconstructions suggest that the Danakil microplate started to rotate in Oligocene-Miocene when Arabia was already separated from Africa". This timing is strange. According to several kinematic models (including the ones proposed by me) and to geological evidence, rifting between Arabia and Nubia started between 30 and 27 Ma (early Oligocene), while the Danakil and Sinai microplates formed during the Langhian (~14 Ma) by strain partitioning.

- We apologies for the mistake and corrected the text.

R432: "We propose a scenario in which rifting was triggered by a plume-induced plate rotation". This is incorrect. Rifting in the Red Sea and the Gulf of Aden was triggered by far-field forces as any other process of continental breakup, although the presence of a mantle plume has certainly exerted some influence on the formation of a triple junction and the separation of Somalia from Nubia.

- There is no controversy, this is the point. The "plume-induced plate rotation" mechanism suggest that far field forces are increased by the plume-push. Still, the far field forces are the one that derive riffing.

Comments by Derek Keir

Line 76 - should upper lithosphere be lower lithosphere? The plume impacts the lower lithosphere first.

- We changed the text.

Line 95 - earth should be upper case Earth

- We changed the text.

Line 108 - 114 and section 7.3 - i thought a weakness of the work is lack of discussion of quite a bit of geodynamic modelling work on a similar topic to the aim of this discussion paper.

This manuscript has avoided discussing the various models by Stamps for the region in which GNSS, topography and lithosphere and asthenosphere imaging data have been used to guide numerical simulations to isolate driving forces of extension in NE Africa. These works generally found the gravitational potential energy (GPE) from uplift is a major driver of extension, with base of lithosphere traction rather minimally involved.

For example see Stamps et al., 2014 -
https://agupubs.onlinelibrary.wiley.com/doi/10.1002/2013JB010717

From what i can see the work by Stamps is somewhat in contradiction to the interpretations being made in the new discussion manuscript. This is fine - but i think the text deserves dealing with this in a more convincing fashion.

- We added a discussion in section 7.3

Line 141 - 152 - there is a fair body of literature in Afar and rifted margins more generally that discusses the concept that magnetic stripes could form as a result of magma intrusion and volcanism before the continental lithosphere is fully split - ie magnetic striping on the continnet ocean transition, and subtly prior to full seafloor spreading. This is potentially important for interpretting the onset of seafloor spreading. See section 5.1 of Ebinger et al., 2017 - especially the last paragraph of this and references therein https://agupubs.onlinelibrary.wiley.com/doi/10.1002/2017TC004526

- We modified the text.

line 223 - replace "last" with "the early"

- We changed to "late"

Line 247 - Please double check this is a catalog of Quaternary volcanoes , rather than just Holocene volcanoes . Also google earth should be Google Earth

- Thanks, we used the Holocene catalog from Smithsonian.

Section 6 - consider using the term "escarpment" rather than sharp cliff / cliff. Escarpment is the more globally used term for these topographically prominent rifted margins.

- We changed the text.

End of section 7.1 - i saw this same point mentioned in one of the other reviews. The concept that strain in Afar is commonly somewhat localised in distinct rift segments but which are set within quite a broad strain field influenced by all the plate motions is not new. e.g. Keir et al., 2010, Tectonics identified a SE-NW striking dike intrusion event in the MER that coupled with structural geology and focal mechanisms, amongst other things, was used to interpret that strain from the NE motion of Arabia occurs in the MER of southern Afar. Also see Doubre et al., 2017 GJI and Pagli et al., 2019 which provide good evidence from geodesy for broad extension in central Afar. See Maestrelli et al., 2022 Tectonics for analog models that use a number of model scenarios to reconstruct potential evolution and distribution of extension and faulting of Afar. These models invoke somewhat broad zones in which extension from the various plates interact, and within which strain has a higher gradient (more localised) in some zones.

- We added a discussion in Section 7.1

---

## Author Response (AR2)

23/4/24

Dear Editor,

Please see our revised manuscript "Rift and plume: a discussion on active and passive rifting mechanisms in the Afro-Arabian rift based on synthesis of geophysical data".

We carefully considered the comments and suggestions of the editor and reviewers. We carefully followed and addressed al of the profound comments of the editor Dr. Cadenas, which improved the manuscript significantly. We added some comments in the tracked changes version.

Figures 1, is modified and the order of Fig 5. And Fig. 6 is switched in the revised manuscript.

Hope you will find the revised manuscript suitable for publication in Solid Earth

On behalf all author,

Ran Issachar

---

## Editor Decision (ED2)

- Page 1, line 14: "applying the" instead of using
- Page 1, line 14, maybe add "to the topography and the bathymetry" in between "(…) Gaussians" and "and interpretation (…)".
- Page 1, line 15, maybe add "the" before "interpretation of vertical gravity (…)"
- Page 1, line 16, maybe change "with the aid of these methods" for the applied methods and the published geological observations incorporated.
- Page 1, line 17, morphologies instead of morphology?
- Page 1, line 19, should the verb be in this sentence: "the triple junction (…) developed" or has been developing"?. Because in line 21, "the onset of the triple junction was (…).
- Page 1, line 22, maybe add "," after "by this time".
- Page 1, line 27, maybe add "located to the" between "a nearby pole" and "northwest to the".
- Page 1, line 27, "northwest of the" instead of "northwest to the"?
- Page 1, line 33, maybe change "geophysical data" by "topography, bathymetry and gravity data"
- Page 1, line 33, maybe change "reviewed the available geological data" by a sentence that enumerates the incorporated geological data (volcanoes, ML>4 earthquakes,…).
- Page 2, lines 59-60, "Moreover, detailed compression between changes in plate motions and the 59 activity of plumes, suggests new concepts". This needs a bit of rephrasing.
- Page 2, line 64, maybe you can specify an example for "derived from other case of studies".
- Page 2, line 65, delete the "," in between "and" and "the timing of".
  Page 2, line 67, maybe add "and" after "gravity" and add "and incorporated the distribution of offshore" before "magnetic anomalies", add "ML>4" before earthquakes, add "onshore Quaternary" before "volcano distribution".
- Page 2, lines 67-68, Maybe change ". Using these datasets, we" by "in order to"
- Page 3, line 113, add "," after "in these studies"
- Page 3, line 113, delete "s" in changes.
- Page 3, line 115, maybe add "an" before " abrupt plate speed up"
- Page 3, line 116, maybe add "at" before "~65 Ma"
- Page 3, line 116, add "," before "the acceleration"
- Page 3, line 118, maybe change "was" by "were" and "as capable" by "triggering mechanisms of" and "plate kinematics" by "plate kinematic variations" and remove "even trigger".
- Page 4, line 129, maybe remove "at" and change "distance" by "away"
- Page 4, line 141,  maybe add "." after "(Fig. 3) and "However,"
- Page 4, line 144, maybe change "current" by "active".
- Page 4, line 157, maybe remove "different".
- Page 5, line 158, maybe remove "location".
- Page 5, line 178, add "the" before "dynamic topography component".
- Page 5, line 181, maybe change "present" by "active".
- Page 5, line 191, maybe add something like "to have occurred" between "is estimated" and "between 20 and 18 Ma)".
- Page 5, line 194, maybe change " developed" by "recognized".
- Page 6, line 201, maybe remove "longitude".
- Page 6, line 204, maybe change "our best knowledge" by "in the literature".
- Page 6, line 206, maybe you can remove "the" before rifting
- Page 6, line 208, add "at" before "~23Ma".

- Page 6, line 212, maybe remove "latitudes" and add "N" after "16º" and "18º".
- Page 6, line 213, maybe remove "latitude" and add "N" after "22º".
- Page 6, add "the" before "opening rate".
- Page 6, line 222, maybe change "the onset of" by "age of".
- Page 6, line 229, maybe change "north" by "northwards".
- Page 6, line 229, maybe add "." before "However".
- Page 6, line 234, maybe change "at different times" by "diachronically" and "when" by "with the onset of".
- Page 6, line 235, maybe change "started" by "occurring".
- Page 7, line 258, change "above" by ">".
- Page 7, line 268, maybe remove "where in Fig. 5" and add "(Fig. 5)" after "grey colours".
- Page 8, line 282, maybe change "less straight" by "irregular"
- Page 8, line 290, maybe change "less sharp" by "smoother" or "gentler"
- Page 8, line 296, maybe change "with" by "within"
- Page 8, line 300, maybe add another figure as reference, because figure 3 does not include BGA and VGG data.
- Page 8, line 304, change "westward" by "westwards".
- Page 8, line 204, maybe change "meeting the" by "running"
- Page 8, line 313, maybe a "to" is missing here because of rewriting and edition.
- Page 8, line 313, maybe add "," after "rift axis".
- Page 8, line 316, maybe change "faulted bathymetry" for "fault activity inferred from the bathymetry".
- Page 9, line 321, maybe add "which" before "steeply"
- Page 9, line 321, maybe add "," after "along the basin".
- Page 9, line 323, maybe you should remove "-" before "1700" if you use deep afterwards.
- Page 9, line 326, maybe remove "intensive" before "seismic activity"
- Page 9, line 331, maybe change "more than" by ">".
- Page 9, line 351, maybe change "bays" for "arcuate rift segments".
- Page 9, line 353, maybe add "rift" before "axial segments".
- Page 10, line 364, maybe change " authors to suggest that this region reflects" for "to the interpretation of this region as"
- Page 10, line 373, maybe add "." before "However".
- Page 10, line 377, maybe add "mainly" after "supported".
- Page 10, younging trend of magmatic??
- Page 10, line 378, maybe remove "and other arguments".
- Page 10, line 388, maybe change "which are spread over a" for "covering"
- Page 11, line 407, maybe add "entirely" before "dictated"
- Page 11, line 426, maybe rephrased to something like " a central BGA high and lateral VGG peaks"
- Page 11, line 431, maybe change "in" by "during"
- Page 11, line 439, maybe change "suggested" by "suggest"
- Page 11, line 440, maybe change "marks" by "recorded".
- Page 12, line 470, remove "between Africa and Arabia" at the end of the sentence, because it is repeated.
- Page 12, line 502, remove "already" and change "showed" by "show"
- Page 12, line 506, remove "ed" from "calculated"
- Page 12, line 519, maybe change "manner" by "tectonic scenario".
- Page 12, line 526, add "s" to "divide".

- Page 12, line 527, maybe change "a synthesis of" by "the"
- Page 12, lien 527, maybe add "and" before "gravity" and add "and the integration of published" before "magnetic anomalies (…)".
- Page 12, line 530, maybe add "," before "in contrast to".
- Page 12, line 532, maybe add "Rift" after "Main Ethiopian".
- Page 12, line 538, maybe change "from which the rift develops" by "development".
-

---

## Author Response (AR3)

7/5/24

Dear Editor,

We are very happy that you find our paper suitable for publication in Solid Earth. Please see our corrections following the topic editor's file, we carefully considered all the corrections.

On behalf all author,

Ran Issachar